# The circadian clock and darkness control natural competence in cyanobacteria

Arnaud Taton [1], Christian Erikson [1,3], Yiling Yang [2,4], Benjamin E. Rubin [1,3], Scott A. Rifkin [1], James W. Golden [1] & Susan S. Golden [1,2✉]

The cyanobacterium *Synechococcus elongatus* is a model organism for the study of circadian rhythms. It is naturally competent for transformation—that is, it takes up DNA from the environment, but the underlying mechanisms are unclear. Here, we use a genome-wide screen to identify genes required for natural transformation in *S. elongatus*, including genes encoding a conserved Type IV pilus, genes known to be associated with competence in other bacteria, and others. Pilus biogenesis occurs daily in the morning, while natural transformation is maximal when the onset of darkness coincides with the dusk circadian peak. Thus, the competence state in cyanobacteria is regulated by the circadian clock and can adapt to seasonal changes of day length.

---

[1] Division of Biological Sciences, University of California, San Diego, La Jolla, CA 92093, USA. [2] Center for Circadian Biology, University of California, San Diego, La Jolla, CA 92093, USA. [3] Present address: Department of Molecular & Cell Biology, University of California, Berkeley, Berkeley, CA 94720, USA. [4] Present address: Institute of Hydrobiology, the Chinese Academy of Sciences, Wuchang District, Wuhan, Hubei Province, China. ✉email: sgolden@ucsd.edu

Horizontal exchange of genetic material through conjugation, phage transduction, and natural competence shaped the evolution of the prokaryotic domain of life[1]. The phenomenon of natural competence was famously discovered by Griffith in 1928 in the Gram-positive bacterium *Streptococcus pneumoniae*[2]. In Gram-negative bacteria such as *Vibrio cholerae* the core apparatus of DNA uptake is the type IVa pilus machine (T4PM) and an assemblage of competence-associated proteins[3]. The T4PM is composed of an extracellular pilus fiber (T4P) and a cell envelope–spanning complex called the basal body[4]. The major pilin PilA, processed by the prepilin peptidase PilD, makes a pilus that extends and retracts through the addition and removal of pilin subunits at the pilus base, powered by the assembly and disassembly adenosine triphosphatases PilB and PilT, respectively. The exogenous DNA (eDNA) that enters the periplasm through the T4PM[5] is pulled in by ComEA[6], a DNA binding protein, and crosses the inner membrane through the ComEC[7] channel. Once the eDNA, processed to single stranded DNA (ssDNA), reaches the cytoplasm, Ssb and DprA bind to the ssDNA and protect it from degradation; DprA also recruits RecA, which recombines the ssDNA with the chromosome[8,9]. Although common mechanisms of DNA uptake and processing are thought to be shared by most transformable bacteria, numerous questions remain unanswered and the regulatory mechanisms that control competence vary among transformable species[10,11].

*Synechococcus elongatus* PCC 7942 is a model cyanobacterium for the study of circadian rhythms[12] and a platform strain for the production of biochemicals[13]. *S. elongatus* has been known to be naturally transformable since 1970[14], but the mechanisms by which it or any other cyanobacterium takes up DNA from the environment are still poorly understood[15–17]. Cyanobacteria constitute a diverse phylum of prokaryotes that established oxygenic photosynthesis at least 3 billion years ago[18] and contribute a large fraction of primary production in the oceans[19]. In contrast to other bacteria, cyanobacteria rely on light as a primary source of energy; accordingly, their cellular activities respond strongly to the presence and absence of light[20]. To synchronize their activities over a diel (24-h day–night) cycle, cyanobacteria use an endogenous circadian clock[12]. The proteins KaiA, KaiB, and KaiC comprise the core oscillator of the clock, which programs a ~24 h cycle marked by the phosphorylation state of KaiC, whose autokinase and phosphatase activities are modulated by KaiA and KaiB. To synchronize the oscillator with the Earth's diel cycle, input pathways that involve KaiA, KaiC, and the histidine kinase CikA monitor the light availability through cellular redox states and energy status[21,22]. To regulate circadian processes, an output pathway relays KaiC phosphorylation states to a two-component system with two antagonistic histidine kinases, CikA and SasA, and the response regulator RpaA[23]. RpaA serves as a master transcription factor that binds more than 134 transcript targets[24] and enables complex circadian gene expression patterns by regulating a cascade of interdependent sigma factors[25].

The cyanobacterial clock is known to regulate global gene expression[26], chromosome compaction[27], cell division[28,29], and metabolite partitioning[20]. Here we screen a dense library of *S. elongatus* randomly barcoded transposon mutants (RB-TnSeq)[30] to uncover the genetic basis and regulation of natural competence in *S. elongatus*. Our data show that competence is controlled by the circadian clock and follows a model of external coincidence, wherein the external stimulus, darkness, must fall at the time of the dusk circadian peak to maximize transformation, and that the regulation of this process adapts to the seasonal change of day length.

## Results

**The machinery.** In order to globally identify all genes that enable natural competence, we transformed an *S. elongatus* RB-TnSeq library[30,31] with eDNA carrying selectable markers that recombine into the *S. elongatus* chromosome at neutral sites (NS)[32]. We reasoned that transposon mutants that are incompetent for transformation would be selectively missing from the transformed population. Mutant colonies were collected as pools from selective and non-selective control conditions and the barcodes were sequenced. The relative abundance of mutants for each represented gene in both conditions was interpreted as a measure of fitness with a corresponding level of confidence (T-value) for each locus[31] (Fig. 1a, Supplementary Data 1). Using fitness values above or below +1 and −1 and an absolute T-value above 4, we identified with high confidence (false discovery rate < 0.001) 12 genes whose loss of function positively affect natural transformation and 47 genes whose loss of function negatively affect natural transformation. Several functional categories that negatively or positively affect natural transformation were represented. Genes whose loss strongly improves transformation encode homologs of Argonaute and Cas4. Strikingly, nearly all genes that encode the T4PM are necessary for cells to take up DNA (Fig. 1a, b), experimentally revealing that, as for other phyla of bacteria, natural competence in *S. elongatus* is mediated by the T4PM and specialized competence proteins. Competence-specific factors include ComEA, ComEC, ComF, and DprA, which are implicated in binding, pulling, and processing exogenous dsDNA as ssDNA in the periplasm, and translocating it to the cytoplasm. The RB-TnSeq data were confirmed by transformation assays performed on selected knockout mutants (Fig. 1c).

Most of the identified proteins are conserved among cyanobacteria (Fig. 2a), even though only a few species are known to be naturally competent. Several novel genes are required for transformation in *S. elongatus* but not widely conserved among cyanobacteria. These include two pairs of genes, *pilA3* and Synpcc7942_2591, and Synpcc7942_2486 and Synpcc7942_2485, whose products carry a type IV pilin-like signal peptide (Fig. 2a, b). While neither PilA3 nor Synpcc7942_2591 is encoded in other naturally competent cyanobacterial species, transformation assays demonstrated that both genes are required for transformation in *S. elongatus* (Fig. 2c). In addition, PilA3 overexpression leads to increased transformation efficiency (Supplementary Fig. 1). Sequence similarities, a characteristic prokaryotic N-terminal methylation motif, and structural predictions clearly identified PilA3 as a pilin subunit. The Synpcc7942_2591 protein has no relatives with a known function but carries a PilW T4P assembly domain and so we designated it PilW. The roles of Synpcc7942_2486 and Synpcc7942_2485 are enigmatic; orthologs for Synpcc7942_2486 were found in competent strains but orthologs for Synpcc7942_2485 were not. Transformation assays showed that each gene is required for natural transformation of *S. elongatus* and we named them *rntA* and *rntB* (Fig. 2c). Transmission electron microscopy showed that strains missing *pilA3* and *pilW* or *rntA* and *rntB* are still piliated (Supplementary Fig. 2).

**Dusk and dark.** Competence is regulated and transient for most bacteria that are naturally transformable[10], but no strict competence regulation has been described for *S. elongatus*; however, transformation efficiency increases with the duration of exposure to eDNA and is enhanced in darkness[15]. We hypothesized that natural competence is regulated by the circadian clock, which globally controls gene expression and metabolism in this organism[20,26].

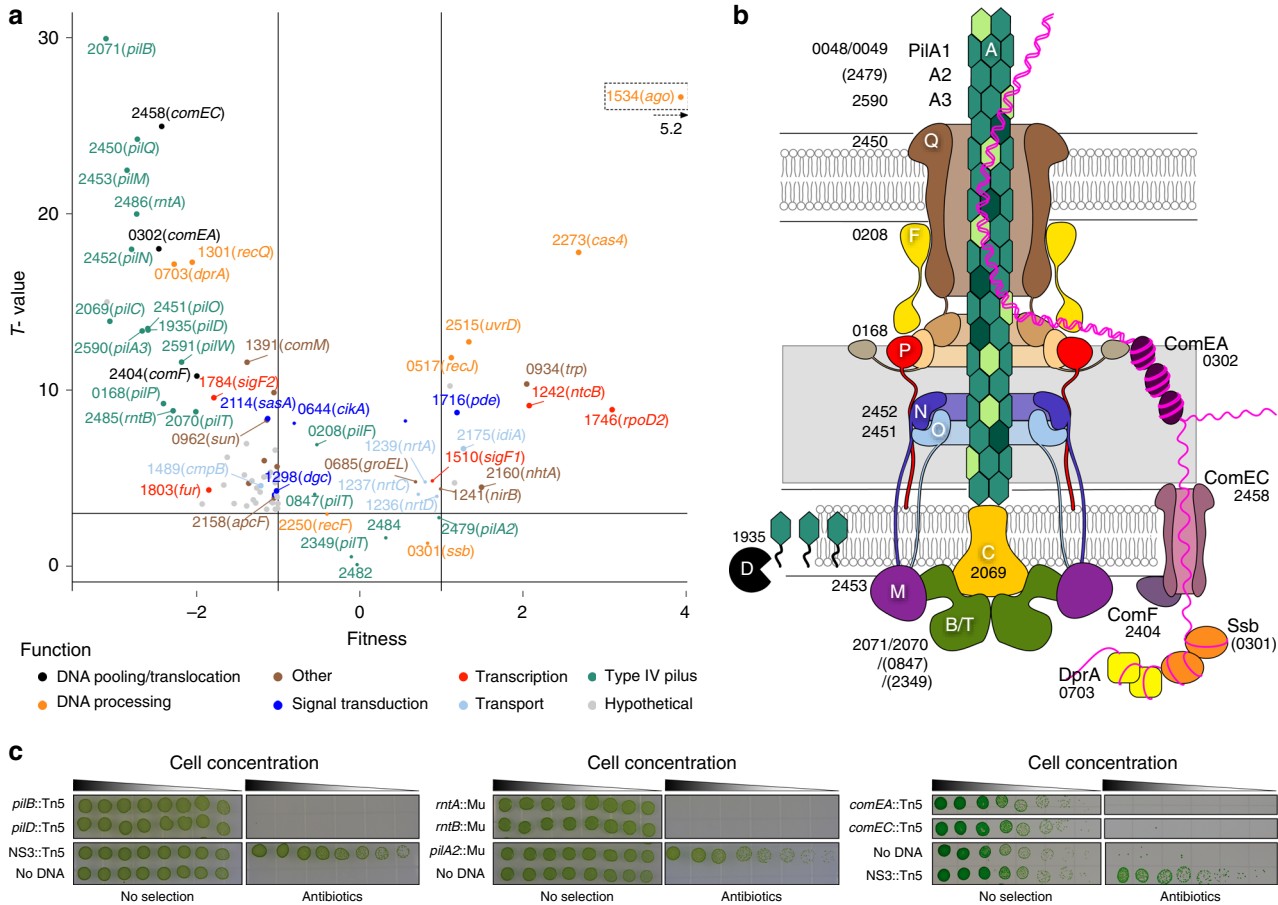

**Fig. 1 T4PM and competence proteins comprise the machinery for natural competence in cyanobacteria. a** Fitness values and confidence levels associated with barcoded transposon mutants for natural transformation. Loci that are essential for or strongly contribute to natural transformation are characterized by a low fitness with a high T-value, whereas improved transformation following knockout of a locus generates a high fitness with a high T-value. Fitness values were calculated for 1885 loci, from three independent experiments, representing a total of 82,495 distinct mutants obtained under selective conditions and 90,872 under control conditions. With the exception of a few selected loci representing selected functional categories, only those whose loss of function had a strong and significant fitness effect are displayed on this plot. The data points were labeled with *S. elongatus* locus tag (Synpcc7942_) numbers and gene names in parentheses. **b** Depiction of the natural competence machinery, based on the T4PM architectural model of Chang et al.[4], in which each protein is labeled with *S. elongatus* locus tag number. T4PM or competence protein homologs that are not required for natural transformation are in parentheses. **c** Transformation assays performed on insertion knockout mutants of selected loci. Strains of *S. elongatus* that carry a chromosomal insertion in loci known to not affect natural competence, *pilA2* or neutral site 3 (NS3), served as positive controls for transformation, and those same strains to which no eDNA was added served as negative controls. The assays were performed with two independent clones for each strain and yielded similar results. Source data are provided as a Source Data file.

The RB-TnSeq library screen did not reveal clock genes among the highest determinants of natural competence. The RB-TnSeq data for clock histidine kinase genes *sasA* and *cikA* had mild negative fitness values, but knockout mutants for either of these genes are still naturally transformable[33]. Loss of the master regulator RpaA has pleiotropic effects, including extreme sensitivity to darkness[34]; as a result, this locus is so poorly represented in the control library that its loss of competence is not reflected in the pipeline that compares the transformed and untransformed barcodes, as was also true for elucidation of genes required for growth in light-dark cycles[35]. To delve further, we analyzed available circadian and light/dark transcriptomes for *S. elongatus*[24,36,37] (Supplementary Data 1). The data clearly indicate that most of the T4PM is circadian controlled and expressed in the morning. In contrast, a subset of genes required for natural transformation, including *pilA3*, *pilW*, *rntA*, and *rntB*, are circadian with highest expression levels at dusk (Fig. 3a) and are induced by darkness (Fig. 3b).

Electron microscopy of cultures grown in a 12-h light/12-h dark cycle (LD 12:12) confirmed that pili biogenesis occurs largely in the morning (Fig. 3c, Supplementary Fig. 3). Denuded cells incubated from Zeitgeber Time (ZT), the time elapsed since the lights were turned ON, of ZT 0 (dawn) to ZT 6 (middle of the day) display many pili at the cell surface, while denuded cells incubated during the second half of the night from ZT 18 to 0 (dawn) remained bald, and cells incubated at the day–night transition from ZT 9 to 15 carried only a few pili. To experimentally test whether natural competence is circadian and dark-induced, quantitative transformation assays on circadian-entrained (LD 12:12) cultures of *S. elongatus* were performed on the second day following their release into constant light (LL). Circadian time (CT) is the biological internal time of entrained cells kept in constant light with CT 0 and CT 12 being subjective dawn and dusk, respectively. For each CT sample, the cells were incubated with eDNA either in the light or in the dark (Fig. 3d and Supplementary Fig. 4). Transformation efficiency peaked at dusk and remained high through the middle of the

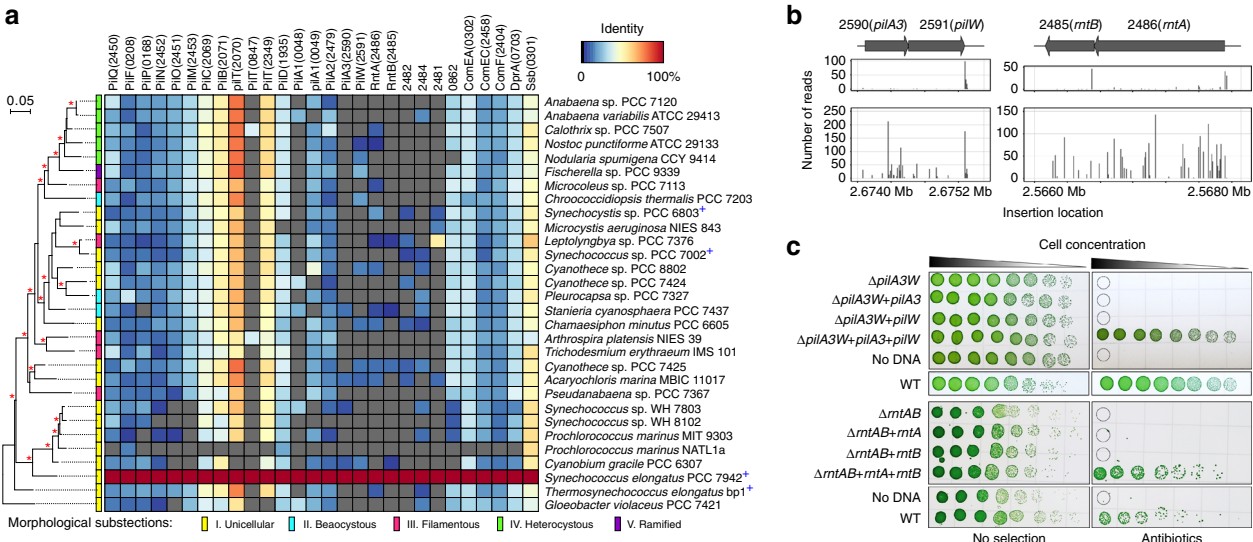

**Fig. 2 The *S. elongatus* T4PM contains minor pilins required for natural competence. a** Heatmap of protein sequence homologies between T4PM and competence proteins of *S. elongatus* and other strains of cyanobacteria. Gray cells indicate no homolog in the corresponding strain. Strains organized according to 16S rRNA gene phylogeny, inferred by Maximum Likelihood. Nodes supported by a bootstrap ≥ 70% (*n* = 500) are marked with a red asterisk and strains that are known to be naturally competent are marked with a blue plus sign. **b** Genomic organization and RB-TnSeq insertion loci for two sets of genes essential for transformation. For each these coding regions and flanking sequences, the number of sequencing reads for insertion-mutant barcodes in selective (top boxes) and non-selective control (bottom boxes) conditions is shown for one representative experiment. **c** Transformation assays performed on deletion and complemented strains of the *pilA3* and *pilW*, and *rntA* and *rntB* loci. For each locus, both open reading frames were deleted then complemented separately and together. WT *S. elongatus* PCC 7942 served as a positive control and the fully complemented strains to which no DNA was added served as negative controls. The assays were performed with three independent clones for each strain and yielded identical results. Source data are provided as a Source Data file.

subjective night. From CT 20 to CT 5, incubation with eDNA yielded no or few transformants. Strikingly, peak transformation efficiency begins at CT 11: the circadian peak for specific essential competence genes, the time when cells anticipate a transition from day to night, and also when darkness starts to strongly impact transformation efficiency.

We predicted that clock-mediated expression of certain dusk-peaking, dark-induced genes is central for the regulation of competence. Accordingly, semi-quantitative transformation assays at key CT points of a *kaiBC*-null mutant compared to the wild-type (WT) strain showed lower efficiencies that remained constant at all time points; the complementation of *kaiBC* partially restored the daily rhythm of competence with a loss of transformation efficiency at CT 22 (Supplementary Fig. 5a). To more definitively investigate the role of the circadian oscillator, we performed quantitative transformation assays at key CT points with two mutants that express phosphomimetic alleles of *kaiC* that lock the oscillator in distinct time-of-day-specific phosphorylation states: a dephosphorylation state more abundant before dawn (KaiC-ET) and a phosphorylation state more abundant at dusk (KaiC-SE)[38]. The expression of *kaiC-ET* (dawn) as the only *kaiC* allele resulted in a strain that lost competence entirely (Supplementary Fig. 5b). The *kaiC-SE* (dusk) strain had transformation levels that increased by orders of magnitude when incubated in the dark (~25–500×), achieving high levels of transformation at all time points (Fig. 4a). This finding is consistent with induction of competence accompanying the circadian physiology of dusk and early night (Fig. 3d). These results confirm that KaiC orchestrates the timing of natural competence and indicate that the dusk phosphorylation state of KaiC potentiates competence.

While the induction of dusk genes, including the competence genes *pilA3* and *pilW*, is under the control of phosphorylated RpaA (Supplementary Data 1)[24], the mechanism behind

increased natural competence in response to darkness is less clear. The expression of hundreds of circadian-regulated genes is affected by changes in light intensity, implicating a network of transcription regulators that is still not well understood[37]. The RB-TnSeq screen revealed the antagonistic importance of two sigma factors in natural competence: *sigF2* and *rpoD2* (Fig. 1a). The role of RpoD2 appears complex and not strictly tied to natural competence[39,40]. SigF2, however, is required for natural transformation (Figs. 1a, 4b), has circadian dusk-peaking expression that increases with darkness (Fig. 3a, b), and is a target for RpaA and RpaB[24,37], which are both implicated in the regulation of circadian genes by light availability. SigF2 is paralogous to SigF, which in other cyanobacteria is associated with pili formation and motility[41]. Although, SigF is also tied to the T4PM in *S. elongatus* based on previous RB-TnSeq screens[42], those screens did not reveal a role for SigF2. Thus, we asked whether SigF2 contributes to the dark induction of natural competence. The relative transcript levels of selected genes required for transformation, and (except *pilQ*) that have been reported to be dark-induced were investigated by RT-qPCR for the WT, a *sigF2* knockout, and a complemented *sigF2* strain (Fig. 4c). To determine whether key genes are induced by darkness, LD-entrained cells were either maintained in the light or transferred to darkness at ZT 12 and sampled for RNA 2 h later, which corresponds to the peak of natural competence in LD. For the WT, our results sampled 2 h after dusk generally agreed with data sampled 3–4 h before dusk[37] that *sigF2, pilA3, rntA*, and *dprA* are dark-induced. However, *comEA* expression decreased in the dark after dusk, whereas it increased in the dark before dusk[37], suggesting that *comEA* dark induction is tightly controlled by the clock. Notably, *dprA* induction did not occur in the absence of SigF2, consistent with evidence for *sigF2*-dependent expression of *dprA* in a recent transcriptomics dataset obtained in LL[25]. The complementation of *sigF2* from a

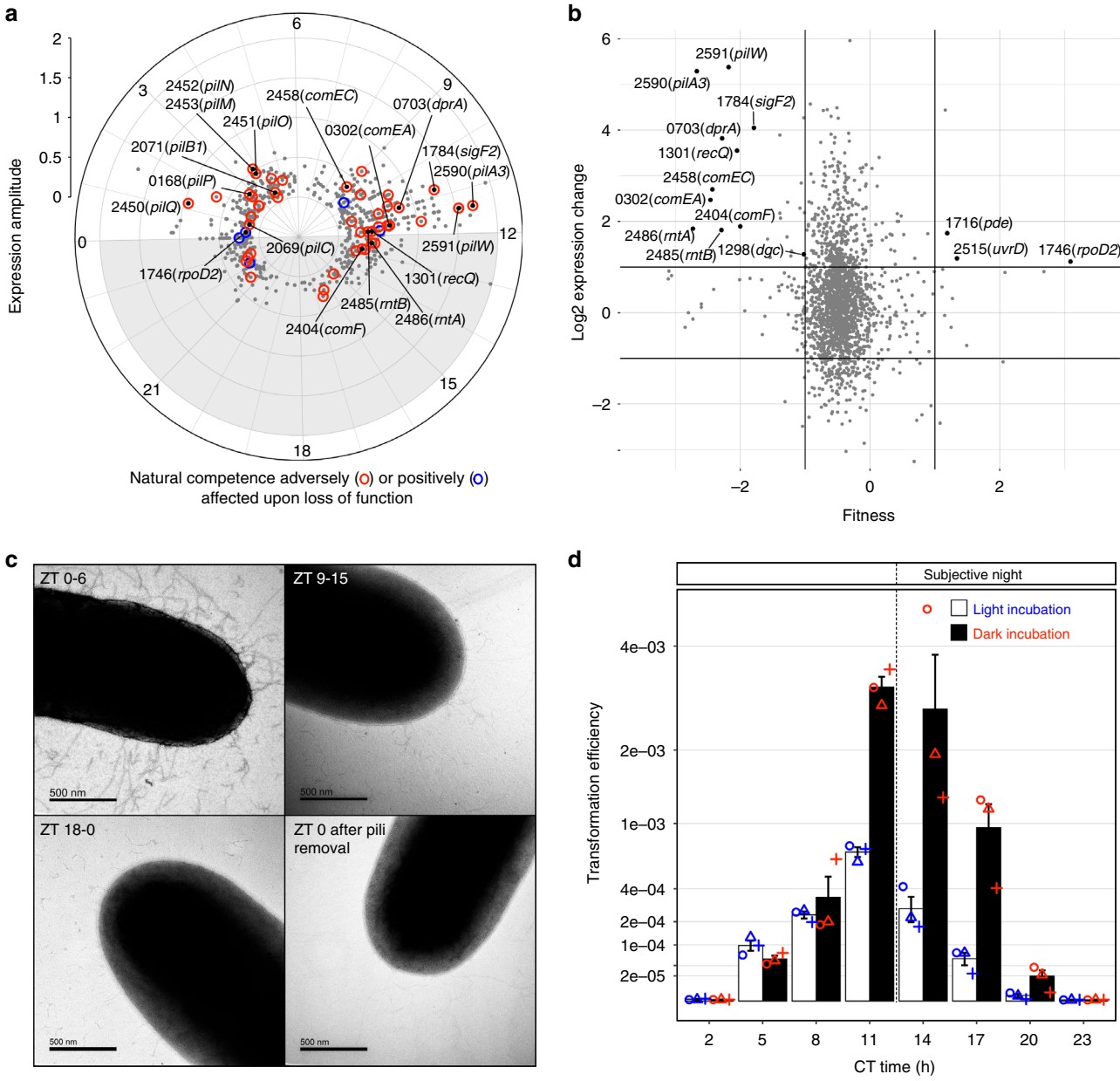

**Fig. 3 Natural competence is under circadian control and induced in the dark. a** Amplitude of circadian-expressed genes reported at peak time, calculated from *S. elongatus* circadian transcriptomics[24,36]. Most of the T4PM genes (including *pilB, M, N, O, P, Q*) are circadian controlled and expressed in the morning while a subset of genes needed for natural transformation such as *pilA3, pilW*, and *rntA* and *rntB*, and the competence genes including *comEA, comEC, comF*, and *dprA*, are circadian with highest expression levels at dusk. A red circle indicates a gene whose loss of function adversely affects natural competence and a blue circle indicates a gene whose loss of function positively affects natural competence. **b** Expression change for genes after a 60-min shade pulse given 8 h after the onset of the day[37], plotted according to RB-TnSeq fitness values as described for Fig. 1. **c** Transmission electron micrographs of cells grown in LD and incubated at different time points (Zeitgeber time, ZT time) for 6 h after removal of their pili. Electron micrographs were chosen as representatives of 5–10 pictures of cells selected randomly and prepared from two biologically independent cultures for each time point. **d** Transformation efficiency over a 24-h circadian cycle. Entrained cells of WT *S. elongatus* released in LL were incubated with eDNA at different circadian time points (CT time). Efficiencies were calculated as the number of antibiotic-resistant colonies per colony forming unit (CFU) without selection upon transformation of three biologically independent cultures (circles, triangles, and plus signs) and plotted as mean values ± standard error of means (SEM) on a square-root scale to compress large values. A similar experiment using a circadian reporter strain of *S. elongatus* and incubated with eDNA carrying a different antibiotic-resistance gene that integrates at a different neutral site yielded similar results (Supplementary Fig. 4). Source data are provided as a Source Data file.

constitutive promoter restored *dprA* dark induction and resulted in overexpression of *dprA*. The downregulation of PilQ at dusk in darkness supports the idea that fewer T4PM are produced in early night, while specific competence-related T4P are made at this time. The results suggest that *sigF2*, which responds to light availability[37], participates in a complex regulatory network that

mediates dark-induced competence at dusk, and implicates *dprA* as an important target (Fig. 4c).

**Seasonal photoperiod.** The regulation of competence by the clock and darkness is reminiscent of a model of external

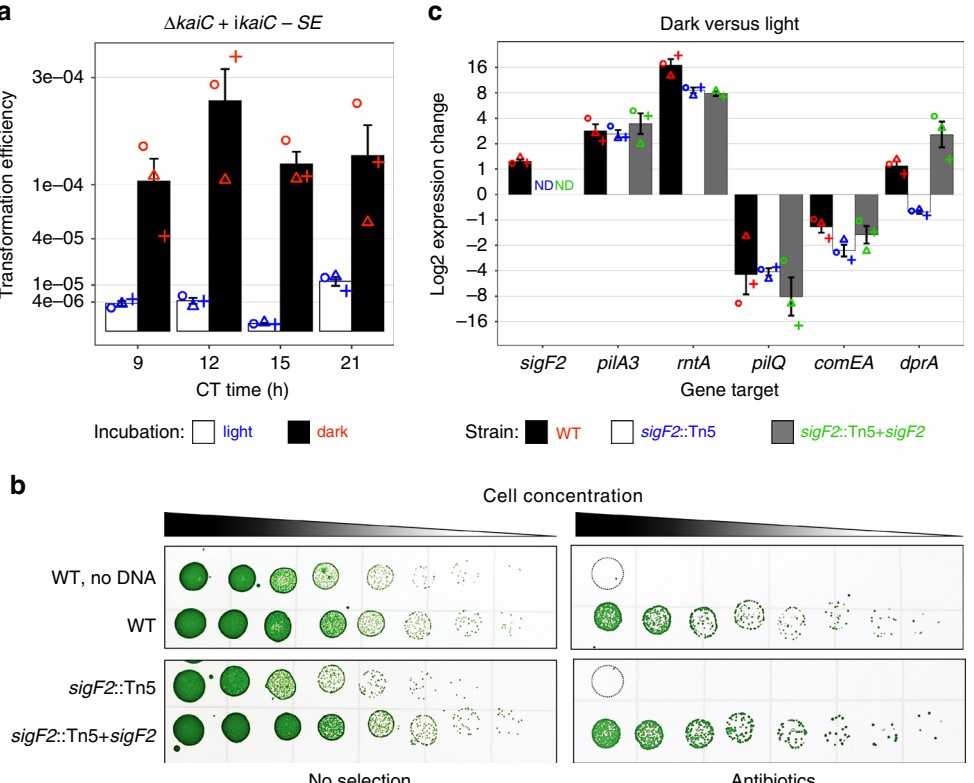

**Fig. 4 The timing of natural competence is controlled by the phosphorylation state of KaiC and requires SigF2-dependent expression of DprA.**
**a** Transformation efficiency at 4 time points over a 24-h circadian cycle of Δ*kaiC* complemented with a mutant *kaiC-SE* phosphomimetic allele that mimics the dusk phosphorylation state of KaiC. Efficiencies were calculated as the number of antibiotic-resistant colonies per CFU without selection upon transformation of three biologically independent cultures (circles, triangles, and plus signs) and plotted as mean values ± SEM on a square-root scale. **b** Transformation assays performed on knockout and complemented strains of *sigF2*. WT *S. elongatus* PCC 7942 with and without added eDNA served as controls. The assay was performed using three independent clones for each strain, which yielded similar results. **c** Expression analysis of selected T4PM and competence genes in a *sigF2* knockout strain, a *sigF2* complemented strain, and a WT control measured by RT-qPCR on cultures collected 2 h after circadian dusk with cells that went into the dark at ZT 12 or were maintained in light. Fold-changes were calculated as $2^{-\Delta\Delta Ct}$ in dark relative to light conditions for three biologically independent cultures (circles, triangles, and plus signs) of each strain and plotted as mean values ± SEM. The expression of *sigF2* by RT-qPCR was not determined (ND) in the *sigF2* knockout and *sigF2* complemented strains. Source data are provided as a Source Data file.

coincidence proposed by Erwin Bünning in 1936[43]. This model explains how the underlying circadian cycle adjusts to different seasonal photoperiods to change the abundance at dusk or dawn of components whose activity is directly or indirectly affected by the presence of light. For example, external coincidence underlies the timing of daily hypocotyl elongation in *Arabidopsis*[44]. Although cyanobacterial growth varies over seasons in the environment[45], no studies address a contribution of photoperiodic regulation or a role for the internal circadian clock. We asked whether the daily pattern of natural competence is affected by changes in photoperiod that mark seasonal variations.

Levels of transformation were measured every 2 h over a 24-h time course in LL and LD using cultures of *S. elongatus* entrained with distinct photoperiods: LD 12:12 for regular days and 16:8 for long days associated with summer; cells did not thrive in an LD 8:16 winter photoperiod. To maximize transformation levels in LL, incubations with eDNA were performed in darkness; while in LD, incubations were performed in either light or dark according to the diel condition at the time of transformation. The results of this experiment (Fig. 5) agreed with those presented in Fig. 3d and with the prior finding that the clock tracks midday[46]. In LL (Fig. 5a, c), the peak and trough times of transformation occur 2 h later in long days vs. regular days, whereas the onset of night occurs 4 h later. Thus, peak transformation efficiency occurs at the onset

of subjective night in regular days and 2 h prior to subjective night in long days. By extrapolating these data over a few cycles, the mfourfit method[47] also calculated a shift of 2.1 h between regular and long days. In regular days, the amplitude of transformation efficiency was greater than in long days, falling off sharply on either side of the peak, whereas a broad peak of transformation in long days resulted in high competence during a time window of 8 h vs. only 4 h in regular days. In LD (Fig. 5b, d), a dramatic difference in transformation occurs between day and night. In regular days, competence peaked to almost 5% of cells being transformed during the night but was almost entirely suppressed from dawn until dusk. In long days, a small fraction of the cells remains competent at dawn. The peak of transformation occurs 2–4 h after nightfall in both regular and long days. In long days, the peak time is delayed by 4 h compared to regular days. To further illustrate the cooperative effect of darkness and circadian control on natural competence, additional assays were performed on cultures maintained for 4 h in the light after anticipated nightfall and in the dark at the onset of the day (Fig. 5b, d, white bars below or next to black bars). In both photoperiods, natural competence remains low after anticipated nightfall if the cells are maintained in light conditions, demonstrating the need for dark induction; furthermore, as in LL, darkness has little effect on natural competence if not aligned with the circadian peak.

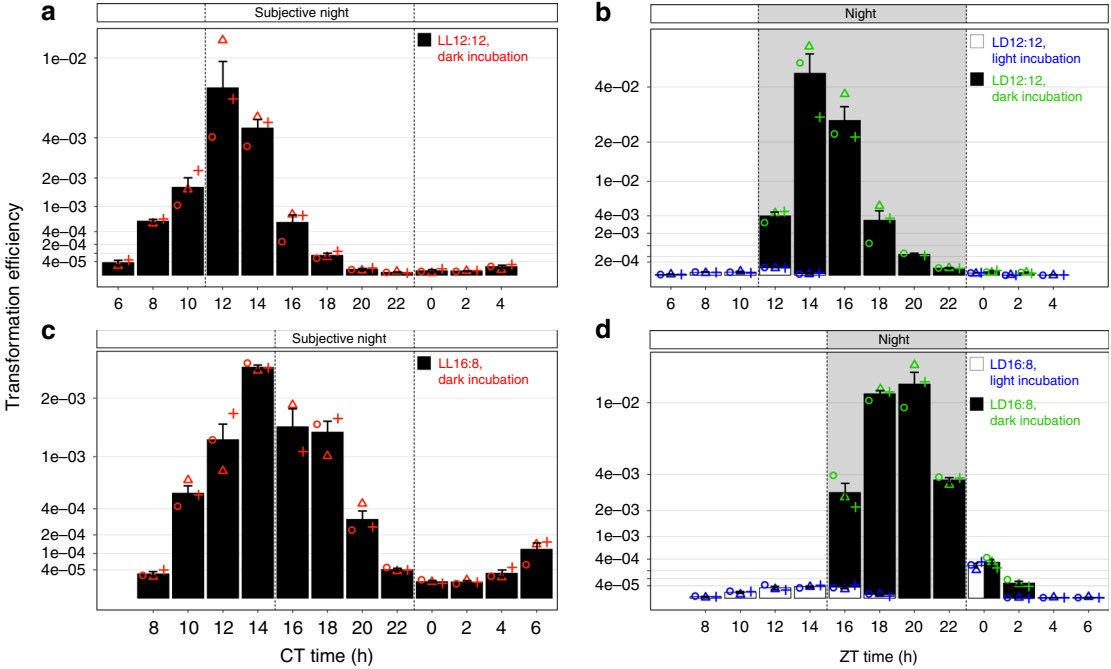

**Fig. 5 Natural competence is under circadian control, requires external coincidence with darkness, and responds to changes in day length.**
Transformation efficiency over a 24-h circadian cycle of cultures entrained with (**a**, **b**) 12-h day length, and (**c**, **d**) 16-h day length under (**a**, **c**) constant light, and (**b**, **d**) light-dark conditions. Efficiencies were calculated as the number of antibiotic-resistant colonies per CFU without selection upon transformation of three biologically independent cultures (circles, triangles, and plus signs) and plotted as mean values ± SEM on a square-root scale. White and black bars are either overlaid or shown side-by-side to make both values visible. Source data are provided as a Source Data file.

## Discussion

Cyanobacteria are important photosynthetic primary producers in many environments[19] and have great potential as platforms for the production of renewable biochemicals[13], so understanding their genetic exchange mechanisms is of ecological and practical importance. In-depth studies of natural competence have largely focused on a few heterotrophic bacteria[3,48–50], but little is known about the process in most prokaryotic phyla. In cyanobacteria, some of the conserved T4PM and competence proteins had been shown to be essential for transformation[16,51]. Our work experimentally extends this knowledge to encompass all proteins that are required for the process including those that could not be identified solely by bioinformatics[17]. Beyond conserved T4PM and competence proteins, we also identified hypothetical proteins and PilA3, a rare minor pilin, which are required for natural transformation of *S. elongatus*. Similarly, minor pilins were found to be necessary for floc formation in the cyanobacterium *Syne-chocystis* sp. PCC 6803[52].

The findings explain the value of two empirically derived steps in the standard natural transformation protocol for *S. elongatus*: washing the cells by resuspension in fresh medium, and incubating them with DNA in darkness[15]. We found that the washing step denudes cells of pili, most of which are unlikely to participate in transformation because they are synthesized when cells are not in a competent state, and may form a physical network (Supplementary Fig. 3) that interferes with DNA access to a small number of transformation-competent pili. The dark incubation is clearly important for inducing components that are required for the transformation process.

Although a fundamental blueprint for DNA uptake and processing may be conserved among bacteria, the regulatory mechanisms and inducing cues vary among phyla[10,11]. The finding of a daily rhythm of natural competence, controlled by the circadian clock and nightfall, makes physiological sense for the phototrophic cyanobacteria, invoking recurrent general

themes of nutritional status and stressful conditions[10] as drivers of competence. Their internal clock allows cyanobacteria to anticipate the daily change in light and dark, which is important as they rely upon light to drive photosynthetic metabolism and growth, and must adapt their physiology daily to darkness[34]. While the circadian rhythmicity of competence can be tied directly to the phosphorylation state of KaiC and the RpaA-dependent expression of dusk-peaking genes[24,26] that are essential for natural competence, the network of transcriptional regulators beyond RpaA and RpaB[37] is unmapped. We found that the alternative sigma factor SigF2, which is a target of RpaA and RpaB[24,37], factors into the dark-inducing mechanism, likely through its role in the expression of DprA, which loads ssDNA on RecA prior to recombination[3,10].

A convergence of environmental signals and internal circadian rhythms is a hallmark of photoperiodic flowering in plants[53] and seasonal reproductive rhythms in mammals[54], and controls daily growth rhythms in *Arabidopsis*[44]. Similarly, the circadian peaking of gene expression required for natural transformation is synergistic with darkness in rendering cyanobacterial cells competent. The fitness benefit of circadian-regulated transformation is not obvious, but some hypotheses can be proposed. Competence is regulated among most naturally transformable bacteria[10], suggesting that there is a genetic or physiological load associated with constitutive transformation. In *S. elongatus* the clock is a master regulator that controls transcription and metabolism globally[12], so its role as the gatekeeper of competence is not surprising. The specificity for early night as the competence window, and the clock's ability to adjust the circadian cycle to coincide well with dusk in different photoperiods, point to a fitness advantage related to this phase of the cycle. One possibility is that potentially threatening phage DNA may be more abundant in the environment during the day[55]. The evolutionary drive for bacteria to coexist with phages using a variety of mechanisms is well documented[56]. It is also possible that the permissive window for

transformation is the only time during the daily cycle that internalized DNA can be incorporated into the genome. The circadian clock controls numerous processes in *S. elongatus* including cell division[12]. According to the prevailing model, the clock selectively inhibits cell division during a temporal window during the night, and cells divide freely at other times of the day[28,29]. A recent study further showed that instead of solely applying an open-close gate on division, cells must integrate internal and external cues to decide when to divide[57]. The evolutionary pressure for circadian gating of cell division and competence may be related, as the timing of maximal transformation roughly corresponds to the window during which cell division is disallowed. Growth arrest coregulated with competence has been observed in other model organisms[11]. Moreover, the *S. elongatus* chromosome undergoes a clock-dependent compaction cycle, with chromosomes maximally compacted around dusk and decompacting during the night[27]. Thus, there may be a limited time during the day when eDNA taken up from the environment could be safely and productively incorporated into the genome by homologous recombination.

Why did it take so long to discover the circadian rhythm of competence in *S. elongatus*, which is famous for its robust clock? The strain is so compliant as a model organism that daily variations were not notable until queried rigorously[15]. For convenience cultures are grown in LL, where they remain rhythmic, but the population exhibits variations in clock phasing such that competent cells are always present during routine laboratory growth, and the transformation of such cultures always leads to more transformants than are required for most genetic experiments.

## Methods

**Strains, plasmids, and molecular methods.** *Escherichia coli* strains were grown at 37 °C in lysogeny broth (LB, Lennox) liquid culture or on agar plates, supplemented as needed with: 100 µg ml$^{-1}$ ampicillin (Ap), 20 µg ml$^{-1}$ spectinomycin (Sp) plus 20 µg ml$^{-1}$ streptomycin (Sm), 50 µg ml$^{-1}$ spectinomycin (Sp), 15 µg ml$^{-1}$ gentamycin (Gm), 17 µg ml$^{-1}$ chloramphenicol (Cm), 50 µg ml$^{-1}$ kanamycin (Km), and 50 µg ml$^{-1}$ nourseothricin (Nt). Unless noted otherwise, *S. elongatus* PCC 7942 and its derivative strains were grown in BG-11 medium[58] as liquid cultures with continuous shaking (125 rpm) or on agar plates (40 ml, 1.5% agarose) at 30 °C under continuous illumination of 200 µmol photons m$^{-2}$ s$^{-1}$ from fluorescent cool white bulbs. Culture media for recombinant cyanobacterial strains were supplemented as needed with 2 µg ml$^{-1}$ Sp plus 2 µg ml$^{-1}$ Sm, 2 µg ml$^{-1}$ Gm, 7.5 µg ml$^{-1}$ Cm, 5 µg ml$^{-1}$ Km, and 5 µg ml$^{-1}$ Nt.

Recombinant strains of *S. elongatus* were constructed by natural transformation using standard protocols[32]. Briefly, liquid cultures were grown to an OD$_{750}$ of 0.5–0.6. Cells were pelleted by centrifugation at 4500 × *g*, washed once with BG-11 medium and once with 10 mM NaCl, then resuspended in BG-11 to a concentration of 1 × 10$^9$ cells ml$^{-1}$. Samples (200 µl) were incubated in darkness for 16 h with 500–1000 ng of plasmid DNA. Transformation reactions were then plated on BG-11 agar with appropriate antibiotics and incubated for 4–5 days under continuous illumination until isolated colonies appeared. Complementation and overexpression strains were constructed by expressing a gene(s) ectopically in *S. elongatus* chromosomal neutral sites: NS1, NS2, and NS3[32,59]. To complement genes essential for natural transformation, such as *pilA3* and *pilW*, *rntA* and *rntB*, or *sigF2*, we first constructed strains with one or both genes expressed ectopically and then deleted the native loci. Complete segregation of the deleted loci was PCR-verified, and three independent clones were picked as biological replicates for subsequent experiments. Plasmids were constructed using the GeneArt Seamless Cloning and Assembly Kit (Life Technologies) and propagated in *E. coli* DH5α or TOP10 with appropriate antibiotics. Gene knockouts in *S. elongatus* were constructed using *S. elongatus* UGS library plasmids[60] or with plasmids engineered with CYANO-VECTOR assembly[61]. Strains and plasmids used in this study are listed in Supplementary Table 1.

**RB-TnSeq library screen for natural competence genes.** To screen the *S. elongatus* RB-TnSeq library[30] for genes involved in natural competence, an aliquot of the library archived at −80 °C was thawed at 37 °C for 2 min in a water bath, then inoculated into three flasks of 100-ml BG-11 with Km and incubated at 30 °C under continuous illumination of 30 µmol photons m$^{-2}$ s$^{-1}$ for 1 day. Then the cultures were placed on an orbital shaker at 150 rpm under 70 µmol photons m$^{-2}$ s$^{-1}$ of continuous light until they reached an OD$_{750}$ of ∼0.3. A sample of the library prior to transformation (T$_0$) was collected to determine the population baseline.

Then natural transformations were performed as described above except that cells were incubated with 1 µg of exogenous plasmid DNA in low light (10 µmol photons m$^{-2}$ s$^{-1}$) instead of complete darkness to preserve mutants that do not survive light-dark cycles. For each experiment, 10 transformation reactions were prepared and plated on selective (Nt or Sp+Sm) and non-selective (control condition) BG-11 agar plates and incubated for 4–5 days under continuous illumination until isolated colonies appeared. Three experiments were performed using plasmid DNA carrying either an Sp/Sm resistance gene (pAM5329) that recombines into the *S. elongatus* chromosome at NS1 (performed twice) or an Nt resistance gene (pAM5544) that recombines at NS2[32]. For each experiment, 70,000–250,000 colonies were collected, pooled, and stored at −80 °C for genomic DNA extraction[32]. Then the barcodes were amplified, sequenced and quantified in perl 5v18 using previously developed BarSeq protocols[62]. The barcodes were curated to keep only barcodes located within the middle 80% of each coding sequence, and to keep only the genes represented by at least three barcodes in different positions and with at least 15 T$_0$ reads across replicates[31,35]. This curation resulted in 82,495 barcodes (also named strains) for the experimental conditions, 90,997 barcodes for the controls, and 99,872 barcodes for the T$_0$ samples distributed across 1885 genes. For each gene, a fitness value that describes how the loss of function affects natural competence and a corresponding statistic were estimated in R version 3.6.0 using methods and scripts developed previously[31,35]. The method uses maximum likelihood to fit a pair of nested linear mixed effects models to the sample- and read-normalized log-2 transformed counts:

$$y_{i,j,k} = \mu_g + C_j + B_i + \varepsilon_{i,j,k}; B_i \sim iid\, N(0, \varsigma_g^2); \varepsilon_{i,j,k} \sim iid\, N(0, \sigma_g^2) \qquad (1)$$

$$y_{i,j,k} = \mu_g + B_i + \varepsilon_{i,j,k}; B_i \sim iid\, N(0, \varsigma_g^2); \varepsilon_{i,j,k} \sim iid\, N(0, \sigma_g^2) \qquad (2)$$

where $y_{i,j,k}$ is the normalized log-2 value for barcode $i$ in gene $g$ in condition $j$ for sample $k$, $\mu_g$ is the average value for the gene, $C_j$ is the fixed effect of the condition $j$, $B_i$ is a random effect for barcode $i$, and $\varepsilon_{i,j,k}$ is the residual. To account for the population baseline in the library prior to transformation (T$_0$), the log-2 count for a barcode from the starting pool is subtracted from the values for experimental and control conditions. To prevent errors when log-transforming the read counts, a pseudocount of one is added to the number of reads for a barcode for each sample. For each gene, the significance of the fitness difference between experimental and control conditions was determined by comparing the difference in the −2log likelihoods of the models to the right tail of a chi-square distribution with one degree of freedom, estimating a *p*-value, and accounting for multiple testing by controlling the false discovery rate[63] with a significance cutoff of 0.001. If the experimental and control conditions differentially affect the prevalence of a barcode, then including the condition-specific term $C_j$ should significantly improve the model fit. The raw data and the annotated R scripts to perform this analysis are provided as Supplementary Data 2.

**Heatmap of protein homologies and 16S rRNA gene phylogeny.** Proteins that belong to T4PM including minor pilins, that carry a pilin-like signal peptide, or are recognized as essential for natural competence in other bacteria were initially identified in *S. elongatus* based on its genome annotation, the available literature[17,64], or sequence homologies using proteins with known functions from other organisms. For each protein encoded by *S. elongatus*, BLASTp searches were conducted against selected cyanobacterial strains using BioBIKE (http://biobike.csbc.vcu.edu/)[65] to identify the best reciprocal hit with an e-value ≤ 0.001. The heatmap was then generated based on the sequence identities using the heatmap.plus R package.

A multiple alignment of 16S rRNA gene sequences and the phylogenetic tree were constructed using the software package MEGA7[66]. The alignment was generated with MUSCLE[67] implemented in MEGA7 using default parameters. Positions that can be used reliably in a phylogenetic analysis were extracted with Gblocks 0.91b[68] at settings that allowed the most relaxed selection of blocks and covered 1338 positions of the alignment. Evolutionary history was inferred by using the Maximum Likelihood method based on the Kimura 2-parameter model[69] implemented in MEGA7. The tree with the highest log likelihood (−9284.19) is shown. The percentage of trees in which the associated taxa clustered together was calculated for each node and indicated with an "*" when above 70%. The tree is drawn to scale, with branch lengths measured in the number of substitutions per site.

**Determination of the peak expression time.** The time of peak expression for each circadian-regulated gene in Fig. 3a was estimated from the cosine wave $Y(t) = A\cos[2\pi(t - \varphi)/P]$, where $A$ is the expression amplitude, $P$ is the period and $\varphi$ is the phase adjusted to $0 \ge \varphi < 24$, of a cyclic gene as determined using published circadian transcriptomics[36].

**Transformation assays of non-synchronized cells.** Knockout, complementation, and overexpression strains for selected genes identified as important for natural competence, as well as appropriate control strains, were grown as described earlier and subjected to semi-quantitative transformation assays. Transformation reactions (cells incubated with eDNA) were prepared as described above using the subject strain and appropriate plasmid DNA (pAM5328, pAM5544, or pAM5554)

that carries an antibiotic-resistance gene, and that will recombine into a neutral site. For each strain, assays were performed with two or three independent clones. Transformation reactions were serially diluted and spotted (8 spots of 4.5 µl) on agar plates (10 cm square petri dishes) without antibiotic to determine total colony forming units (CFU), and with antibiotic to select for transformed clones.

**Transformation assays of light/dark entrained cells**. To obtain quantitative transformation efficiency levels over a circadian time course, our transformation protocol was modified to account for the time sensitivity of these experiments and the effect of light vs. dark incubation with eDNA.

For the experiments presented in Fig. 3d and Supplementary Fig. 4, precultures of *S. elongatus* entrained in 12-h light/12-h dark cycles (LD 12:12) were used to inoculate triplicate 100-ml cultures to an $OD_{750}$ of 0.1 in BG-11 medium at the onset of light. They were grown using stirred flasks for 2 days in LD 12:12 and then transferred to constant light (LL) on the third day. On the second day in LL, quantitative transformation assays were performed every 3 h using 200 µl of cells, washed once with BG-11, concentrated to $1 \times 10^9$ cells ml$^{-1}$ and incubated for 3 h with 500 ng of plasmid DNA either in the light or in the dark. The incubations in the dark were limited to 2 h followed by 1 h in the light to reduce the chance of resetting circadian phase. Transformation reactions were stopped by serially diluting and inoculating samples (25 µl) onto selective or non-selective BG-11 in 12-well plates (3 ml/well, 1.5% agar). The plates were incubated for 5 days at 30 °C in LL until colonies had appeared. For each plate, wells with isolated colonies were photographed (typically 2 wells for the selective medium and 1 well for the control). Pictures were processed with ImageJ and curated manually to produce binary images without background and where adjacent colonies were separated. The number of colonies was determined with ImageJ. Transformation efficiencies were calculated as the number of antibiotic-resistant colonies per CFU without selection. This experiment was first performed with WT *S. elongatus* using pAM5544, which carries a Nt resistance gene and integrates at NS2 by homologous recombination (Fig. 3d). This experiment was later repeated with a circadian reporter strain, AMC1300, using pAM5328, which carries a Gm resistance gene and recombines at NS3 (Supplementary Fig. 4).

For the experiments presented in Fig. 4, with the exception of AMC2109, which is dark sensitive and was maintained in LL, precultures grown in LD 12:12 were used to inoculate 100-ml cultures to an $OD_{750}$ of 0.1 in 250-ml Falcon tissue culture flasks bubbled with air (0.1 L min$^{-1}$). Cultures were grown in LD 12:12 for 3 days until they reached an $OD_{750}$ of ~0.5. On the third day at nightfall, cells were concentrated to $5 \times 10^8$ cells ml$^{-1}$ and distributed as 200-µl aliquots in a white 96-well plate with a clear bottom. The plate was placed in the incubator between two light sources to minimize shading effects at the bottom of the wells and to maintain a light intensity of 200 µmol photons m$^{-2}$ s$^{-1}$ on both sides of the plate. At that time, the light sources were turned OFF until dawn, 12 h later. Afterward, the 96-well plates were maintained in LL and quantitative transformation assays were performed at four time points. For each time point, 200 µl of plasmid DNA was added to the wells, the cells and eDNA were mixed by slowly pipetting up and down, and samples were incubated in the dark for 1 h by covering the top and bottom of the wells with opaque aluminum sealing tape. After incubation, transformation reactions were serially diluted, inoculated into 12-well plates, incubated for several days, photographed, and processed to determine colony counts and transformation efficiencies as described above.

For the experiments on day length presented in Fig. 5, precultures of *S. elongatus* grown in LD 12:12 and LD 16:8 were used to inoculate 100-ml cultures to an $OD_{750}$ of 0.1 (LD 12:12) or 0.05 (LD 16:8), which were grown in tissue culture flasks as described above. On the third day at nightfall, the cells were concentrated and distributed in 96-well plates, which were incubated in LD 12:12 or LD 16:8 starting with a dark period of 12 and 8 h for the cells entrained in regular or long days, respectively. Quantitative transformation assays were performed every 2 h, as described for the previous experiment. The time course started at the middle of the day, at ZT 6 (6 h after dawn for cultures entrained in regular days) and ZT 8 for the cultures entrained in long days. Because the incubators were set to constant light, dark conditions were achieved by covering the bottom and top of wells with opaque aluminum sealing tape for 12 h (regular days) or 8 h (long days). Transformation reactions were incubated for 1 h either in the dark or in the light. After incubation, the transformation reactions were serially diluted, inoculated into 12-well plates, and incubated as described above. Triplicate plates were stacked and rotated every day to equalize the total amount of light that each plate received. Wells were photographed and the pictures were processed to determine colony counts as described above. An average of 73 colonies/well were counted for 466 wells. For this experiment, 56 transformations were performed in triplicate (168 transformations in total).

**RT-qPCR**. Precultures grown in LD 12:12 were used to inoculate 100-ml cultures to an $OD_{750}$ of 0.1 in 250-ml Falcon tissue culture flasks bubbled with air (0.1 L min$^{-1}$). Cultures were grown in LD 12:12 for 3 days until they reached an $OD_{750}$ of ~0.5. On the third day at nightfall, each culture was divided into two subcultures with one maintained in constant light and the second put into the dark, and 2 h later the cells were collected, poured over ice in a 50 ml falcon tube and spun down at 4 °C for 10 min at $4500 \times g$. Cell pellets were frozen at −80 °C. RNA

was extracted from triplicate cultures for each strain and condition. Cell pellets were resuspended in 700 µl of TRIzol and incubated for 5 min at 95–100 °C followed by 5 min at room temperature. The mixture was spun down ($16,000 \times g$) for 5 min at 4 °C and the supernatant extracted with 700 µl (1 volume) of 100% EtOH at 4 °C. All following steps were performed at 4 °C using RNase-free tubes and tips. RNA was isolated using the Direct-zol Quick RNA miniprep Kit (Zymo Research) following the manufacturer's instructions. RNA concentration ranged from 900 to 960 ng µl$^{-1}$ as measured with a Nanodrop (Thermo Fisher Scientific). RNA was treated with DNaseI (RNase free, Thermo Fisher Scientific) following the manufacturer's instructions and stored at −80 °C. cDNA was synthesized from 500 ng of RNA template per sample with the High-Capacity RNA-to-cDNA Kit (Applied Biosystems). RT-qPCR reactions were performed in triplicate with the Biotool 2x SYBR Green qPCR master mix (Low ROX) reagents following the instructions in The Applied Biosystems StepOne Real-Time PCR Systems kit. Primer sequences and target genes are listed in Supplementary Table 2. Gene expression fold-changes in cultures incubated in the dark were calculated relative to cultures incubated with light (Fig. 4c) using the $2^{-\Delta\Delta Ct}$ method[70] with *rnpB* (Synpcc7942_R0036) as the reference.

**Electron microscopy**. Small aliquots (200 µl) of *S. elongatus* cultures entrained in LD 12:12 were fixed with 0.5% glutaraldehyde for 5–10 min at room temperature. Sample grids (Formvar/Carbon 100 mesh, Copper) were floated for 2 min on a drop of the fixed cells, rinsed three times by floating the grids on drops of water, and floated on a drop of 1% uranyl acetate for 2 min to stain the cells. Excess uranyl acetate solution was removed with filter paper and grids were air dried for a few minutes before electron microscopy. EM was performed with a JEOL 1200 EX II TEM equipped with a Gatan Orius 600 7-megapixel bottom-mount digital camera (2.7k × 2.7k).

To determine the timing of pili biogenesis, precultures of *S. elongatus* grown in LD 12:12 were used to inoculate 100-ml cultures to an $OD_{750}$ of 0.1 in 250-ml Falcon tissue culture flasks bubbled with air (0.1 L min$^{-1}$), which were grown in LD 12:12 for 3 days until they reached an $OD_{750}$ of ~0.5. On the third day at nightfall, cells were concentrated (200 µl at $10^8$ cells ml$^{-1}$) and distributed in a 96-well plate; the plate was placed in the dark incubator until the next morning. At ZT 0, ZT 9, and ZT 18 the concentrated cells were washed twice with BG-11 to remove their pili, resuspended with 200 µl of BG-11 in the 96-well plate, and returned in the incubator for 6 h until the cells were prepared for electron microscopy. Cells were washed twice using the same procedure as in preparation for transformation. Liquid cultures were spun down by centrifugation at $4500 \times g$ for 3 min and then the cells were vigorously resuspended in 1 ml of BG-11 using a vortex for 10 s. Supplementary Fig. 3 illustrates that this procedure denudes cells of their pili.

**Circadian bioluminescence monitoring**. *S. elongatus* strains expressing P*psbA*-luxCDE and P*kaiBC*-luxAB or P*furA*-luxAB were grown at 30 °C for 2 or 3 cycles in LD 12:12 in the conditions described for the corresponding transformation assays (Supplementary Fig. 4). At the onset of night on the day before being released into constant light, 20 µl of cells were distributed in 96-well agar plates[71]. On the day of the transformation assays, the plates were placed in a TECAN Infinite 200 Pro equipped with a stacker microplate handling system where plates were maintained at 30 °C in LL. Bioluminescence readings were recorded every 3 h for 48 h. Data were analyzed with BioDare2 using linear detrending and normalized to the mean (https://biodare2.ed.ac.uk/)[47].

**Reporting summary**. Further information on research design is available in the Nature Research Reporting Summary linked to this article.

## Data availability
The RB-TnSeq results and previously published transcriptomics analyses underlying Figs. 1a, 2b, 3a, b are provided in the Supplementary Data 1 file. The raw data to perform the RB-TnSeq analysis are provided in the Supplementary Data 2 file. Descriptions of plasmids, strains and RT-qPCR primers are available as supplementary tables; additional supporting materials are provided as supplementary figures. The data underlying Fig. 1a, c, 2a-c, 3a-d, 4a-c, 5a-d, and Supplementary Figs. 4a, b and 5a are provided in the Source Data file. All other data produced and/or analyzed to support the findings of the study are available from the corresponding author upon request.

## Code availability
The computer codes used to process the sequencing reads and quantify the barcodes as well as the computer codes used to obtain fitness values and the corresponding statistics for each strain in experimental vs. control conditions were published previously[31,35,62]. The annotated R scripts to perform the RB-TnSeq analysis are provided in the Supplementary Data 2 file.

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

## Acknowledgements

We thank D. Welkie, C. Sancar, and R. Simkovsky for their input on the experimental design and data analyses, B. McKnight, L.C. Lowe, and C. Peterson for assistance with strain construction and transformation assays. We thank T. Meerlo at the UC San Diego electron microscopy facility and K. Jepsen at UC San Diego IGM sequencing facility for technical support. Funding was provided by the National Institute of General Medical Sciences of the National Institutes of Health under award number R35GM118290 (to S.S.G.) and R01GM118815 (to J.W.G.). The content is solely the responsibility of the authors and does not necessarily represent the official views of the National Institutes of Health. This material is also based upon work supported by the National Science Foundation under Grant number IOS-1754894. Any opinions, findings, and conclusions or recommendations expressed in this material are those of the author(s) and do not necessarily reflect the views of the National Science Foundation.

## Author contributions

A.T., J.W.G, and S.S.G. conceived of and designed the project. A.T. led or performed the experimental work and analyzed the data, C.E. performed the RB-TnSeq transformation screens and helped with strain constructions and preliminary transformation assays. B.E.R. provided the RB-TnSeq library and obtained barcode sequences. Y.Y. performed RT-qPCR experiments. S.A.R. contributed to RB-Tn-Seq analytic tool. A.T., J.W.G., and S.S.G. wrote the paper, which was reviewed and edited by all authors.

## Competing interests

The authors declare no competing interests.

## Additional information

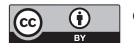

