## [Peer Review File · Nature Communications]

Reviewers' comments:

Reviewer #1 (Remarks to the Author):

In this manuscript the authors investigate the genetic basis of natural transformation in the cyanobacteria *Synechococcus elongatus* and how competence is regulated. Competence for natural transformation has been poorly studied in cyanobacteria and this work fills a major knowledge gap by exhaustively identifying the genes required for transformation. Although, most were already known in other models because this system is highly conserved. Yet the authors identified genes specific to *S. elongatus*. In addition, it establishes very clearly that competence is under control of the circadian clock, suggesting that regulation of competence is deeply connected to the cell's physiology.

I found the experiments to be well-controlled, the manuscript is clearly written and the data support the conclusions. I have no major concerns on this manuscript.

Some minor issues could be addressed to improve the manuscript. Listed below in order of appearance in the manuscript:

1. Page 3. Line 8-13. Please insert references relative to the activities of the pilus, ComEA, ComEC and DprA.

For instance:

Role of pilus retraction in transformation: DOI: 10.1038/s41564-018-0174-y

Role of ComEA: DOI: 10.1371/journal.pgen.1004066 and DOI: 10.1046/j.1365-2958.1999.01170.x

ComEC as a channel: DOI: 10.1111/j.1365-2958.2004.04430.x

Role of DprA in DNA protection: DOI: 10.1046/j.1365-2958.2003.03702.x and in stimulating recombination: DOI: 10.1016/j.cell.2007.07.038

2. Since there is no further functional characterization, I see no reason to rename Synpcc7942_2486 and Synpcc7942_2486 as *rntA* and *rntB*.

3. Page 6. Line 14 "genes required for competence". Competence typically refers to the physiological state in which bacteria express the DNA uptake system and can take up DNA. Transformation is the end result. These genes likely play no role in competence, but are required for natural transformation. I recommend changing to "genes required for natural transformation".

4. Page 6. Line 17. Please define or explain "Denuded". Is this relative to the process of removing the pilus as indicated in panel figure Fig3. panel c and in the legend "... after removal of their pili" (page 21 line 2)? If so, please explain how this is achieved. The method section only indicates "washed twice in BG-11 medium" (page 30, line 13). This would remove free pili, but I doubt this is enough to shave off the pili from the cells.

5. Page 7, line 13-14. I see not data relative to this statement.

6. Page 7, line 20-22. "While the induction of dusk genes, including competence genes, is under the control of phosphorylated RpaA (Supplementary Table 1) ...". Based on Supp Table 1, column Q "Expression change in RpaA KO", only *pilA3* expression is affected by RpaA. There are no change in expression for the other competence genes (*comEA*, *comEC*, *dprA*, etc...). In my opinion, this sentence could be removed since no work is performed to assess the role of RpaA.

7. Page 9, line 22. Correct "competence".

8. Page 10, line 13-14. This is not true. While it is true that natural transformation is most often reported for pathogens, one of the best studied model of natural transformation, *Bacillus subtilis*, is a non-pathogenic bacteria.

9. Page 17, figure 1b. In this model, I see PriA (the primosome assembly protein) associated with the ComF and ComEC protein. ComFA, is required for natural transformation in Gram-positive but an homolog is missing in Gram-negative. Blast search typically find PriA as the closest match (because of the conserved helicase domain) but there is no data supporting a role of PriA in DNA uptake (Synpcc7942_0651 is actually missing in the RB-TnSeq data). Please remove PriA from the model.

10. Page 23, figure 5. Please add the photoperiod info to the panels (regular vs long days).

Reviewer #2 (Remarks to the Author):

This manuscript by Taton et al. describes an excellent, and quite original, work on the study of natural competence as influenced by the internal circadian clock and seasonal rhythms of ambient light, in the model cyanobacterium *Synechococcus elongatus*. The authors made use of a clever strategy to select for transformation mutants, and combine specific and global approaches to identify genes involved in the expression of natural competence, including DNA uptake, as well as their regulation by the circadian clock and the duration of the light/dark periods. The outcome is relevant for the understanding of the performance of globally-relevant phototrophic cyanobacteria in natural environments.

I have only found a couple of minor issues to be attended:

-In Fig. 4a. It would be good to include transformation efficiency of the controls: WT and the *kaiC* mutant.

-Fig. 4d, which is mentioned in the last line of page 8, is missing.

Reviewer #3 (Remarks to the Author):

Taton et al. revealed the relationship between natural competence and circadian times using unicellular cyanobacterium *Synechococcus elongatus* which is often used as a model system for prokaryotic circadian clock. Genome-wide screening identified many genes required for natural transformation, including genes encoding subunits of Type IV pilus machine (T4PM). The authors demonstrated that natural competence is under control of circadian rhythms and induced in the dark. Transformation efficiency was altered by genetic manipulation of *kaiC* genes, truly supporting their conclusion. It's also interesting that seasonal photoperiod affected the transformation efficiency.

Undoubtedly, the manuscript is innovative and attractive for broad readers. However, I have two questions to be clarified before acceptance.

1. I agree that natural competency contributed to the evolution of cyanobacteria, but I cannot get the picture really circadian-regulated competency is important for their evolution. In Discussion, the authors hypothesize that growth arrest is coregulated with competence, and I agree it but it sounds just a hypothesis. Please explain in detail the superiority of circadian-regulated competency during evolution of cyanobacteria.

2. Besides *KaiC*, are the mutants of other circadian-related genes (such as *kaiA*, *kaiB*, *rpaA*, and *cikA*) exhibited altered transformation efficiency?

Reviewer #4 (Remarks to the Author):

This manuscript describes a cooperative effect of darkness and the circadian clock on cyanobacteria (*S. elongatus*) natural competence. Authors used an RB-TnSeq library to identify novel genes that are required for *S. elongatus* natural competence. Experiments show that natural competence, in addition to darkness, is regulated by the circadian clock with a peak ~dusk. Several results support this notion: 1) genes required for natural competence are clock-controlled and or dark induced (including the sigma factor sigF2), 2) natural competence depends on the time of the day at which cells are exposed to eDNA in constant conditions (LL) and 3) time of day dependence is lost in clock mutants. While clock regulation of natural competence was observed in both light and dark conditions, higher transformation efficiency was observed (as expected) when cells were in the dark after transformation. A sigma factor, sigF2, was identified as a required regulator of such dark induction of natural competence. Finally, this manuscript shows that the circadian clock regulates seasonal changes in daily oscillations of *S. elongatus* natural competence. The manuscript is overall clearly written, experiments are well designed and conclusions are well supported by experiment results.

Major points:

1) "requirement of sigF2 for natural competence" (page 8 line 2) should be addressed experimentally

Minor comments:

1) Figure 4c. Why sigF2 is not expressed in the sigF2 complemented strain?

2) Reference to Figure 4d in page 8 line 22 should be 4c.

3) Figures 5b and 5d. The text (page 9 lines 13-15) indicates: "in LD, incubations were performed in either light or dark according to the diel condition at the time of transformation", however for some ZT times, results are indicate both light and dark incubation (i.e. ZT 16, 18, 0, 2 in fig 5d).

Response to specific reviewers' comments:

Reviewer #1 (Remarks to the Author):

In this manuscript the authors investigate the genetic basis of natural transformation in the cyanobacteria *Synechococcus elongatus* and how competence is regulated. Competence for natural transformation has been poorly studied in cyanobacteria and this work fills a major knowledge gap by exhaustively identifying the genes required for transformation. Although, most were already known in other models because this system is highly conserved. Yet the authors identified genes specific to *S. elongatus*. In addition, it establishes very clearly that competence is under control of the circadian clock, suggesting that regulation of competence is deeply connected to the cell's physiology.

I found the experiments to be well-controlled, the manuscript is clearly written and the data support the conclusions. I have no major concerns on this manuscript.

Some minor issues could be addressed to improve the manuscript. Listed below in order of appearance in the manuscript:

1. Page 3. Line 8-13. Please insert references relative to the activities of the pilus, ComEA, ComEC and DprA.

For instance:

Role of pilus retraction in transformation: DOI: 10.1038/s41564-018-0174-y

Role of ComEA: DOI: 10.1371/journal.pgen.1004066 and DOI: 10.1046/j.1365-2958.1999.01170.x

ComEC as a channel: DOI: 10.1111/j.1365-2958.2004.04430.x

Role of DprA in DNA protection: DOI: 10.1046/j.1365-2958.2003.03702.x and in stimulating recombination: DOI: 10.1016/j.cell.2007.07.038

Thank you, we added these references as suggested.

2. Since there is no further functional characterization, I see no reason to rename Synpcc7942_2486 and Synpcc7942_2486 as *rntA* and *rntB*.

A long-standing practice in microbiology is to name genes based on the phenotypes of mutants, and this naming often precedes the revelation of biochemical function by many years. We prefer to keep this nomenclature, which is in keeping with the guidelines for the Journal of Bacteriology, as it serves to highlight genes that contribute to the same process, and makes it easier for readers to distinguish loci in the text. Here we showed that the genes carrying the locus tags Synpcc7942_2486 and Synpcc7942_2485 are required for natural transformation and were accordingly given the names *rntA* and *rntB*. We were careful to choose a mnemonic that would not be confused with other known bacterial genes.

3. Page 6. Line 14 "genes required for competence". Competence typically refers to the physiological state in which bacteria express the DNA uptake system and can take up DNA. Transformation is the end result. These genes likely play no role in competence, but are required for natural transformation. I recommend changing to "genes required for natural transformation".

Thank you for this comment. We replaced this instance of "competence" by "natural transformation" or "transformation". Accordingly, we also changed several other instances of "competence" throughout the manuscript with this distinction in mind.

4. Page 6, Line 17. Please define or explain “Denuded”. Is this relative to the process of removing the pilus as indicated in panel figure Fig3, panel c and in the legend “... after removal of their pili” (page 21 line 2)? If so, please explain how this is achieved. The method section only indicates “washed twice in BG-11 medium” (page 30, line 13). This would remove free pili, but I doubt this is enough to shave off the pili from the cells.

The pili of *S. elongatus* PCC 7942 seem quite fragile and are indeed removed by the washing and resuspension steps we use in our transformation protocol. We were surprised, too! We have added a statement in the method section describing how the cells were washed, and added EM pictures of cells collected at ZT 6 that were or were not washed as described (Supplementary Fig. 3).

5. Page 7, line 13-14. I see not data relative to this statement.

Because the transformation performed on the *kaiC*-ET strain did not result in any transformant colonies, we initially decided not to plot these zero data. However, we now provide, as Supplementary Fig. 5b, pictures of a transformation procedure performed on the *KaiC*-ET strain that demonstrates that the strain is not naturally transformable.

6. Page 7, line 20-22. “While the induction of dusk genes, including competence genes, is under the control of phosphorylated RpaA (Supplementary Table 1) ...”. Based on Supp Table 1, column Q “Expression change in RpaA KO”, only *pilA3* expression is affected by RpaA. There are no change in expression for the other competence genes (*comEA*, *comEC*, *dprA*, etc...). In my opinion, this sentence could be removed since no work is performed to assess the role of RpaA.

We agree with the reviewer that the changes in expression for the competence genes other than for *pilA3* and *pilW* may not be significant. Therefore, we modified this sentence according to the reviewer’s comment. We decided not to delete the sentence entirely because it provides an important foundation to understand the circadian regulation of natural competence in *S. elongatus*.

7. Page 9, line 22. Correct “competence”.

This typo was corrected, thank you.

8. Page 10, line 13-14. This is not true. While it is true that natural transformation is most often reported for pathogens, one of the best studied model of natural transformation, *Bacillus subtilis*, is a non-pathogenic bacteria.

Thank you for pointing out this mistake; the sentence has been corrected. We knew better, but had truncated a longer statement to where it was no longer accurate.

9. Page 17, figure 1b. In this model, I see PriA (the primosome assembly protein) associated with the ComF and ComEC protein. ComFA, is required for natural transformation in Gram-positive but an homolog is missing in Gram-negative. Blast search typically find PriA as the closest match (because of the conserved helicase domain) but there is no data supporting a role of PriA in DNA uptake (*Synpcc7942_0651* is actually missing in the RB-TnSeq data). Please remove PriA from the model.

Thank you very much for this insightful note; we have removed PriA from the model.

10. Page 23, figure 5. Please add the photoperiod info to the panels (regular vs long days).

This is a good idea; the photoperiod was added to the panel as requested.

Reviewer #2 (Remarks to the Author):

This manuscript by Taton et al. describes an excellent, and quite original, work on the study of natural competence as influenced by the internal circadian clock and seasonal rhythms of ambient light, in the model cyanobacterium *Synechococcus elongatus*. The authors made use of a clever strategy to select for transformation mutants, and combine specific and global approaches to identify genes involved in the expression of natural competence, including DNA uptake, as well as their regulation by the circadian clock and the duration of the light/dark periods. The outcome is relevant for the understanding of the performance of globally-relevant phototrophic cyanobacteria in natural environments.

I have only found a couple of minor issues to be attended:

-In Fig. 4a. It would be good to include transformation efficiency of the controls: WT and the kaiC mutant.

Transformation assays performed on a WT strain as well as a kaiBC-null and a kaiBC complemented strain are now included as Supplementary Fig. 5a.

-Fig. 4d, which is mentioned in the last line of page 8, is missing.

This label has been corrected.

Reviewer #3 (Remarks to the Author):

Taton et al. revealed the relationship between natural competence and circadian times using unicellular cyanobacterium *Synechococcus elongatus* which is often used as a model system for prokaryotic circadian clock. Genome-wide screening identified many genes required for natural transformation, including genes encoding subunits of Type IV pilus machine (T4PM). The authors demonstrated that natural competence is under control of circadian rhythms and induced in the dark. Transformation efficiency was altered by genetic manipulation of kaiC genes, truly supporting their conclusion. It's also interesting that seasonal photoperiod affected the transformation efficiency.

Undoubtedly, the manuscript is innovative and attractive for broad readers. However, I have two questions to be clarified before acceptance.

1. I agree that natural competency contributed to the evolution of cyanobacteria, but I cannot get the picture really circadian-regulated competency is important for their evolution. In Discussion, the authors hypothesize that growth arrest is coregulated with competence, and I agree it but it sounds just a hypothesis. Please explain in detail the superiority of circadian-regulated competency during evolution of cyanobacteria.

We do not claim that the circadian rhythmicity of competence played an important role in the evolution of cyanobacteria overall, but rather that transformation did. Nonetheless, the reviewer is correct that we had not adequately discussed potential fitness benefits of restricting transformation to the early night. The Discussion has been expanded to propose relevant hypotheses.

2. Besides KaiC, are the mutants of other circadian-related genes (such as kaiA, kaiB, rpaA, and cikA) exhibited altered transformation efficiency?

We have expanded the paragraph on p. 6 that reports the transformability of *sasA* and *cikA* mutants to explain why *rpaA* (which is not transformable) doesn't come through the RB-TnSeq pipeline. Loss of RpaA causes pleiotropic effects which greatly debilitate these mutants if they are not handled very specially, and they are depleted from the library population even under the control conditions. Thus, the relative abundance of *rpaA*-associated bar codes in the transformed population vs the control is very similar (almost none). Likewise, *kaiB* mutants are seriously debilitated if KaiC is present (the logic would require another paper to explain), and generally results in loss of *kaiC* expression because they are in an operon or by selection for suppressors. We added, as supplementary material, transformation assays performed on a *kaiBC*-null strain (which is healthy) and the complemented strain compared to WT at key CT points. We determined that transformation efficiencies in a *kaiBC*-null strain are inferior to those of a WT strain (Supplementary Fig. 5b), but quantification of transformation for other clock gene knockouts were not performed as they do not add value to the story.

Reviewer #4 (Remarks to the Author):

This manuscript describes a cooperative effect of darkness and the circadian clock on cyanobacteria (*S. elongatus*) natural competence. Authors used an RB-TnSeq library to identify novel genes that are required for *S. elongatus* natural competence. Experiments show that natural competence, in addition to darkness, is regulated by the circadian clock with a peak ~dusk. Several results support this notion: 1) genes required for natural competence are clock-controlled and/or dark induced (including the sigma factor *sigF2*), 2) natural competence depends on the time of the day at which cells are exposed to eDNA in constant conditions (LL) and 3) time of day dependence is lost in clock mutants. While clock regulation of natural competence was observed in both light and dark conditions, higher transformation efficiency was observed (as expected) when cells were in the dark after transformation. A sigma factor, *sigF2*, was identified as a required regulator of such dark induction of natural

competence. Finally, this manuscript shows that the circadian clock regulates seasonal changes in daily oscillations of *S. elongatus* natural competence.

The manuscript is overall clearly written, experiments are well designed and conclusions are well supported by experiment results.

Major points:

1) "requirement of *sigF2* for natural competence" (page 8 line 2) should be addressed experimentally

The requirement of *SigF2* was experimentally tested. Transformation assays were performed with a *sigF2*-null strain as well as with a *sigF2* complemented strain. Please see Fig. 4b.

Minor comments:

1) Figure 4c. Why *sigF2* is not expressed in the *sigF2* complemented strain?

The reviewer's comment made us realize that the figure gave the impression that *sigF2* is not expressed in the complemented strain. There is no reason to think that is the case; expression of *sigF2* by RT-qPCR was not determined in the *sigF2*-null and *sigF2* complemented strains. We added a sentence in the figure legend and modified the figure.

2) Reference to Figure 4d in page 8 line 22 should be 4c.

Thank you, we fixed that error.

3) Figures 5b and 5d. The text (page 9 lines 13-15) indicates: "in LD, incubations were performed in either light or dark according to the diel condition at the time of transformation", however for some ZT times, results are indicate both light and dark incubation (i.e. ZT 16, 18, 0, 2 in fig 5d).

The reviewer's interpretation is correct; as part of the same experiment, we performed additional assays where it would be potentially informative, whose results are displayed in figures 5b and 5d, to further illustrate the cooperative effect of darkness and circadian control on natural competence. The experimental design for these assays is explained in the text, later in the paragraph.